# The Intensity of Simulated Grazing Modifies Costs and Benefits of Physiological Integration in a Rhizomatous Clonal Plant

**DOI:** 10.3390/ijerph17082724

**Published:** 2020-04-15

**Authors:** Jushan Liu, Chen Chen, Yao Pan, Yang Zhang, Ying Gao

**Affiliations:** Key Laboratory of Vegetation Ecology, Ministry of Education, Institute of Grassland Science, Northeast Normal University, Changchun 130024, China; liujs606@nenu.edu.cn (J.L.); chenc049@nenu.edu.cn (C.C.); pany317@nenu.edu.cn (Y.P.); zhangy819@jlu.edu.cn (Y.Z.)

**Keywords:** *Leymus chinensis*, physiological integration, compensation growth, cost–benefit, trade-off, clipping

## Abstract

Clonal plants in grasslands are special species with physiological integration which can enhance their ability to tolerate herbivory stress especially in heterogeneous environments. However, little is known about how grazing intensity affects the trade-off between the benefits and costs of physiological integration, and the mechanism by which physiological integration improves compensatory growth in response to herbivory stress. We examined the effects of simulated grazing intensity on compensatory growth and physiological integration in a clonal species *Leymus chinensis* with a greenhouse experiment. This experiment was conducted in a factorial design involving nutrient heterogeneity (high-high, high-low, low-high, low-low), simulated grazing by clipping (0%, 25%, 50% or 75% shoot removal) and rhizome connection (intact versus severed) treatments. Compensatory indexes at 25% and 50% clipping levels were higher than that at 75% clipping level except in low-low nutrient treatments. Physiological integration decreased and increased compensatory indexes when the target-ramets worked as exporter and importer, respectively. Generally, clipping increased both benefits and costs of physiological integration, but its net benefits (benefits minus costs) changed with clipping intensity. Physiological integration optimized compensatory growth at light and moderate clipping intensity, and its net benefits determined the high capacity of compensatory growth. Grassland managements such as grazing or mowing at light and moderate intensity would maximize the profit of physiological integration and improve grassland sustainability.

## 1. Introduction

Clonal plants possess the capacity to share resources, such as carbohydrates, water and nutrients, among interconnected ramets through physiological integration [1,2,3,4]. Physiological integration facilitates the growth and reproduction of clonal plants by providing the ability to share resources among ramets in heterogeneous environments and enhance their ability to tolerate herbivory damage [5,6,7]. However, no study has examined the effects of herbivory intensity on the benefits and costs of physiological integration and how physiological integration improves compensatory growth of clonal plants in response to herbivory.

Physiological integration confers a number of benefits on clonal plants, but involves some costs as well [2,8]. The benefits of physiological integration often overweighed its costs, which is one of the underlying mechanisms of the widespread distribution and dominance of clonal species [9]. Physiological integration provides a plant with a number of benefits, and such physiological integration has been shown to be advantageous for clonal plants growing in spatially heterogeneous environments [10,11,12,13]. It helps plants forage limiting resources, share them among connected ramets, and escape sites where some limiting resource might have been exhausted [14]. On the other hand, clonal growth involves massive turnover of tissues. Such turnover has both energetic and ecological costs because new resources are required to form new tissue and there is uncertainty whether a vegetative offspring will be placed in favorable conditions [15]. Further, both benefits and costs of physiological integration are found to be increased by simulated herbivory [16]. Thus, the balance between benefits and costs seems to vary between species and may also depend on environments, and consequently the response of the trade-offs between benefits and costs of physiological integration to herbivory still needs to be explored.

Clonal plants have higher capacity to tolerate herbivory than other species, and numerous experiments have found that physiological integration promotes compensatory growth of clonal plants [2,17,18]. Plants may respond to herbivory by compensatory growth, which can alleviate the potential negative effects of herbivory [19,20]. The mechanisms of compensatory growth involve changes in physiology and development, as well as the modification of the environment [19]. Compensatory growth following herbivory may result from the stimulation of photosynthesis of remaining green tissues [21,22], reallocation of resources (i.e., carbohydrates, water and nutrients) [23,24] and/or activation of additional meristems because of release of apical dominance [25,26]. However, the magnitude of compensatory growth after herbivory strongly depends on the availability of resources such as nutrients, light and water [27,28]. Plant compensatory growth also varies with herbivory intensities and is maximized at moderate herbivory intensities.

Clonal plants can translocate and share resources through physical integration [12,29]. Such physiological integration can increase the performance of ramets subjected to herbivory [5,30,31]. A few studies have examined the effects of physiological integration on compensatory growth of clonal plants after simulated herbivory [26,32]. Disconnecting the rhizome between intact (undefoliated) and defoliated ramets may strongly retard the recovery or decrease the compensatory growth [33,34]. However, no study has examined the effects of herbivory intensity on the role of physiological integration in determining plant compensatory regrowth.

Here, we report the results of a study that crossed nutrient, clipping and rhizome connection treatments to examine the role of physiological integration in compensatory growth of a clonal species *Leymus chinensis* (Trin.) Tzvel. *L. chinensis* was selected as the study species for this experiment because it is a typical rhizome clonal plant, and is often exposed to grazing and mowing due to its good palatability for livestock [35,36]. Consequently, it is always subject to nutrient heterogeneity and herbivory-induced heterogeneity. Our aims were to answer the following questions: (a) how the intensity of simulated herbivory affects benefits and costs of physiological integration and (b) how physiological integration maximizes the benefits and improves compensatory regrowth at the moderate intensity of simulated herbivory.

## 2. Materials and Methods

### 2.1. Study Site and Plant Species

The study site was located at the grassland ecology research station of Northeast Normal University, Jilin Province, China (44°45′N, 123°45′E, 151 m). The station is located in the southern part of the Songnen Plain in the eastern Eurasian steppe region, which is a semi-arid area with a mean annual precipitation of 350–450 mm and a mean annual temperature ranging from 2.7–4.7 °C.

*Leymus chinensis* (Trin.) Tzvel. is a typical perennial root clone herb distributed in the arid and semi-arid grasslands of Eastern Eurasia. In natural habitats, it applies mainly of sexual and asexual reproduction (rhizome growth). Rhizome growth of this species plays an important role in spatial expansion and colonization [16]. The rhizomes lie horizontally about 5–15 cm under the ground surface, and rhizome length between two adjacent ramets is approximately 2–6 cm under natural field conditions [36,37]. A large number of studies have shown that the clone plant has a strong ability to compensate after grazing [29,38]. It has the characteristics of good cloning integration and plasticity. *L. chinensis* is an important local species used for grazing and recovery of vegetation.

### 2.2. Experimental Design

The experiment was conducted from May to September 2014 in a greenhouse, within a factorial design involving nutrient (high-high, high-low, low-high, low-low), clipping (0%, 25%, 50% or 75% shoot removal) and rhizome connection (intact versus severed) treatments. This experiment included 32 treatment combinations (four nutrient treatments × four clipping treatments × two rhizome-severing treatments) and five replicates for each treatment combination, which resulted in 160 pots in total. Each plastic pot (20 cm in diameter and 15.5 cm deep) were divided into two parts by placing a 1 mm thick plastic partition along the pot midline. Each partition had 1 round hole of 3 mm in diameter at 6 cm from the bottom, which allowed rhizomes of clonal fragments (described below) to pass. The pots were filled with sand that was calcareous and collected from a nearby field. The sand was sieved through a mesh to remove coarse aggregates.

In the field study, uniformly sized *L. chinensis* clonal fragments were selected and carefully excavated from nearby grassland. Each clonal fragment consisted of two ramets interconnected by a rhizome fragment about 5 cm long. Then, 160 clonal fragments were transplanted into pots with a baffle in the middle of the pots, each fragment in one pot. One ramet in each pot was assigned as ‘target-ramet’ and the other ramet on the other side of the baffle was assigned as ‘neighbor-ramet’ [16,18]. Eighty fragments were kept with an intact rhizome through the holes in the baffle, and the rhizomes of the other 80 fragments were severed. After transplantation, the seedlings were watered every day, and half a month after transplanting the seedlings were watered with NH_4_NO_3_ solution at the rate of 15 g/m^2^ and 5 g/m^2^ for the high- and low-nutrient treatments, respectively. Twenty days after being transplanted, target-ramets were subjected to the clipping treatments (0%, 25%, 50% or 75% of shoot height) using a ruler as a reference. The pots were kept in an open greenhouse where the light and temperature were similar to the outside environment. As the greenhouse removed the influence of rain on the plants they were watered daily.

### 2.3. Measurements

Plants were harvested at the end of growing season (170 days after being transplanted), and were separated into leaves, stems and roots. Biomass of each part was measured with an analysis balance (Sartorius, Mettler Toledo), after being dried at 70 °C for 48 h. Total biomass was the sum of all the parts in each pot, and the root shoot ratio (R/S) was calculated by the ratio of root biomass to shoot biomass, the sum of leaf and stem biomass.

### 2.4. Compensation Index Calculation and Physiological Integration Cost–Benefit Analysis

Compensation indexes were calculated by dividing the sum of the final aboveground biomass and the clipped biomass at each clipping level by the final aboveground biomass in the unclipped treatments in the same nutrient and rhizome connection treatments. Costs and benefits of physiological integration were calculated, respectively. Costs were the difference in total biomass between target-ramets in high-high treatments compared to that in high-low nutrient treatments. Benefits was estimated as the increased performance of target-ramets in low-high treatments compared to that in low-low treatments [16,39]. Compensation index and costs and benefits of physiological integration were calculated for each pot.

### 2.5. Statistical Analysis

Three-way ANOVA was used to investigate the effects of clipping treatment, nutrient treatment and physiological integration as fixed factors on total biomass, leaf biomass, shoot biomass and R/S ratio. Tukey–Kramer test was used to examine the difference in total, leaf and stem biomass, root shoot biomass ratio, compensation index, and costs and benefits of physiological integration among clipping treatments for each nutrient and rhizome connection treatment, respectively. It was also used to compare the difference in compensatory index among clipping treatments for each nutrient treatment, and the difference in benefits and costs of physiological integration among clipping treatments. This method was also used to test the difference in compensatory index between high-low and high-high or between low-high and low-low nutrient treatments at each clipping level. Before analysis, data were tested for normality using Levene’s test, and variables were log transformed, where necessary, to meet the assumptions of statistical analyses. All statistical analyses were done with the SPSS 21.0 (SPSS Inc., Chicago, Illinois, US).

## 3. Results

### 3.1. Biomass Accumulation and Allocation

There were significant interactions among physiological integration, clipping and nutrient on R/S, between physiological integration and clipping on leaf biomass and R/S, and between physiological integration and nutrient and between clipping and nutrient on total, leaf and stem biomass (Table 1). In low-high nutrient treatment, heterogeneous with target-ramet as exporter, the ramets with intact rhizome at 25% and 50% clipping levels had higher total biomass than those at other clipping levels, and the ramets with severed rhizome at 50% had higher total biomass than those at other clipping levels (Figure 1b). The ramets with intact rhizome had higher leaf and stem biomass at 25% and 50% clipping levels than those at other clipping levels (Figure 2b and Figure 3b). At 25% clipping level, the ramets with intact rhizome had higher total biomass than those with severed rhizome, and at 25% and 50% clipping levels (Figure 1b) the ramets with intact rhizome had higher leaf biomass than those with severed rhizome (Figure 2b). In other nutrient treatments there was no compensatory growth at 25% or 50% clipping levels.

In high-low nutrient treatment, R/S at 25% and 50% clipping levels were higher than that at other clipping levels (Figure 4a), and in low-high nutrient treatment, R/S decreased with increasing clipping levels (Figure 4b). With increasing clipping levels, R/S of ramets with intact rhizome decreased in high-high nutrient treatment, and R/S of ramets with severed rhizome decreased in low-low nutrient treatment. R/S was higher at the 25% clipping level than that at other clipping levels for ramets with severed rhizome in high-high and with intact rhizome in low-low nutrient treatments, respectively.

### 3.2. Compensation Index and Cost–Benefit of Physiological Integration

Compensation index was lower at 75% clipping levels than that at other clipping levels in high-low, high-high and low-high nutrient treatment (Figure 5). There was no significant difference among clipping levels in low-low nutrient treatment. Ramets had higher compensation index in high-high than that in high-low nutrient treatment at each clipping level. Compensation index was higher in low-high than that in low-low at 25% and 50% clipping levels. The benefits of physiological integration were higher at 25% and 50% clipping levels than that at other clipping levels, whereas the costs were high at 25% and 75% levels (Figure 6).

## 4. Discussion

In this study, clonal plants had higher biomass at 25% and 50% clipping levels in low-high nutrient treatment (Figure 1, Figure 2 and Figure 3), and physiological integration increased and decreased compensation index when the target-ramets worked as importer and exporter, respectively (Figure 5). Clipping increased both benefits and costs of physiological integration, and increased compensatory growth at 25% and 50% clipping levels was dependent on the benefits of physiological integration.

### 4.1. Nutrient and Physiological Integration Affect Plant Response to Herbivory

Results of this study showed that light and moderate clipping increased ramets’ biomass in low-high nutrient treatment, and not in other nutrient treatments, and ramets with intact rhizome had higher biomass than those with severed rhizome (Figure 1, Figure 2, Figure 3 and Figure 4). This suggested that physiological integration enhances compensatory growth of plants when it works as an importer in a heterogeneous environment, which supports our first hypothesis. However, physiological integration had no such effect on plant growth in a homogeneous environment or in a heterogeneous and high-nutrient environment. This suggested that nutrient and physiological integration affected plant response to herbivory.

The increased compensatory growth of a plant as an importer in a heterogeneous environment was consistent with the positive effects of physiological integration in other studies [10,40,41]. The physical mechanism of compensatory growth at light and moderate herbivory intensities usually includes the stimulation of photosynthesis of remaining tissues [21,22], reallocation of resources (i.e., carbohydrates, water and nutrients) [23], and/or activation of meristems from release of apical dominance [25,26]. Transfer of resources between ramets can improve ramet performance when ramets experience different growing conditions, or when some ramets are subject to and others escape local disturbances such as herbivore attacks [42]. When part of a clone is damaged by herbivory, undamaged ramets may aid individual ramets in recovering from herbivory damage by direct translocation of resources, or by mobilization of shared reserves [34,43]. Further, the magnitude of compensatory growth after herbivory strongly depends on the availability of resources [27,28]. So, physiological integration in a heterogeneous environment can promote compensatory growth of the ramets with low nutrient through translating resource from the connected ramets in a high-nutrient environment. The ramets in a homogeneous environment or in a heterogeneous and high-nutrient environment did not affect compensatory growth because in such an environment the target-ramets cannot get benefits from the connected ramets. The dependence of the contribution of physiological integration on the role of the ramets in heterogeneous environments was consistent with another clonal perennial species, *Linaria vulgaris* Mill. [44]. In addition, heave clipping decreased plant biomass even in low-high nutrient treatment in this study, which suggested that the negative effects of heavy herbivory cannot be alleviated by the benefits from the connected ramets in high-nutrient environment.

It was also found in this study that rhizome severing had negative effects on plant growth at 25% and 50% clipping levels, but had no such effects at other clipping levels. This suggested that the contribution of physiological integration on plant growth works at light and moderate herbivory intensities, not at heavy herbivory intensity. The benefits of physiological integration on the growth of ramets growing in stressful conditions or at heavy defoliation have also been shown in various habitats [32,45,46,47]. The benefits of physiological integration at light and moderate clipping levels in this study suggested that at such clipping levels the stimulation of photosynthesis and reallocation of the resources of remaining tissues cannot supply enough resource for regrowth. The translocation of resources from connected ramets supported the compensatory growth of the target-ramets. However, rhizome connection did not increase the performance of target-ramets at 75% clipping levels, which suggested that integration contributed little to the compensatory growth of this species under such conditions. Plants under heavy herbivory need more resource than what the connected ramets can export, and cannot compensate for the biomass loss. So, physiological integration was not involved in plant growth under heavy herbivory.

### 4.2. Trade-off between Benefits and Costs of Physiological Integration-Determined Compensatory Growth

Clipping significantly increased benefits and costs of physiological integration in this study (Figure 6). Physical connection between ramets allows resources to be transported between connected ramets. Its maintenance has commonly been interpreted as costs of physiological integration for the ramets in homogeneous environments, but it confers benefits for the ramets growing under heterogeneous nutrient conditions [44,47,48]. Benefits from physiological integration in heterogeneous environments have been extensively described in several clonal species [10,40,41], but costs in the maintenance inter-ramet connections are always neglected. Benefits of physiological integration on the growth of ramets growing in stressful conditions have also been shown in a number of clonal plant species from various habitats [12,47,49]. However, response of the trade-off between benefits and costs to herbivory has been rarely reported [16]. The increased benefits of clonal integration by clipping in this study indicated that physiological integration increased plant capacity to tolerate herbivory damage. The translocated resource from connected ramets promoted plant growth after herbivory and was more important for plants damaged than for those undamaged by herbivory. Though clonal growth may increase plant fitness, resource allocation to generate new ramets comes at the expense of investment into other tissues more directly associated with resource capture or sexual reproduction [50,51,52]. New clonal fragments also exhibit their own metabolic demands and must be maintained to preserve the clonal network [39,42], which resulted in the metabolic costs of physiological integration. The increased costs of physiological integration by herbivory in this study indicated that physiological integration needed more resource investment to the damaged ramets. So, compensatory growth of damaged ramets enhanced the difference between target-ramets and connected ramets and increased costs of physiological integration [16].

In the present study, benefits of physiological integration were higher at 25% and 50% clipping levels, which contributed to the higher compensation index at such clipping levels. The costs of physiological integration at 25% clipping level were as high as benefits at such a clipping level, but the compensation index at this clipping level was still as high as that at 50% clipping level. These results demonstrated that the benefits of physiological integration determined its contribution to plant compensatory growth following herbivory. Further study is still needed to explore the mechanism of the role of physiological integration in controlling clonal plant response to herbivory.

## 5. Conclusions

Our study demonstrated that light and moderate herbivory enhances plant compensatory growth following herbivory, and physiological integration optimized compensatory growth at light and moderate herbivory intensity. Physiological integration decreased and increased compensatory indexes of clonal plants when the target-ramets worked as exporter and importer, respectively. Further, simulated herbivory (clipping) increased the benefits and costs of physiological integration, and the net benefits (benefits minus costs) of physiological integration determined compensator growth. This study suggested that the trade-off between benefits and costs of physiological integration changed with the intensity of simulated herbivory. Grazing intensity in natural ecosystems might play the important roles in the recovery of clonal species. Grassland managements such as grazing or mowing at light and moderate intensity would maximize the profit of physiological integration and improve grassland sustainability.

## Figures and Tables

**Figure 1 ijerph-17-02724-f001:**
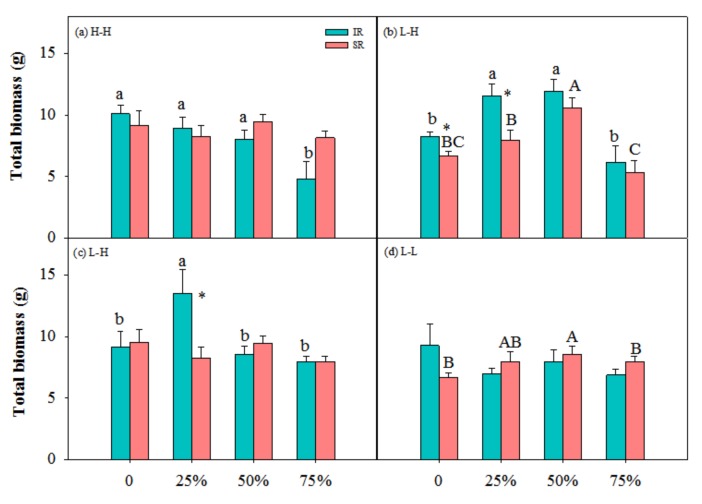
Total biomass of ramets with intact rhizome (IR) and severed rhizome (SR) with increasing clipping levels in different nutrient treatments. (**a**) H-L, (**b**) L-H, (**c**) H-H and (**d**) L-L indicate high-low, low-high, high-high and low-low nutrient treatments, respectively. The error bars represent standard errors (SE), and bars with different lower-case and upper-case letters indicate significant difference among clipping levels for ramets with intact and severed rhizome, respectively (*p* < 0.05). *—significant difference between ramets with IR vs. SR at the same clipping level.

**Figure 2 ijerph-17-02724-f002:**
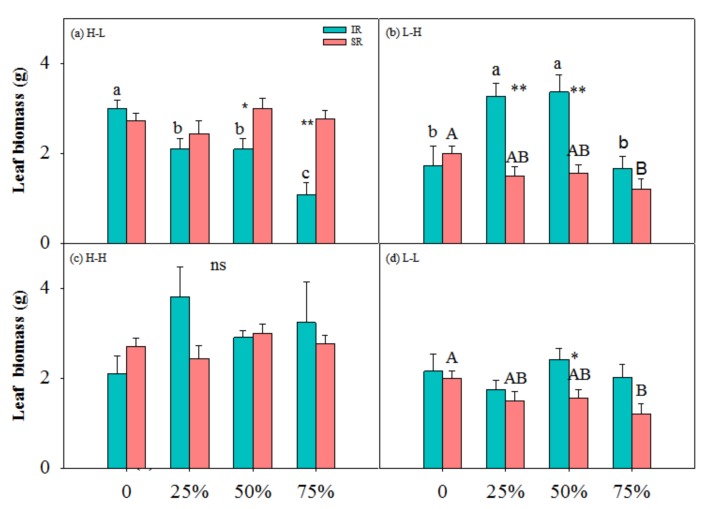
Leaf biomass of ramets with intact rhizome (IR) and severed rhizome (SR) with increasing clipping levels in different nutrient treatments. (**a**) H-L, (**b**) L-H, (**c**) H-H and (**d**) L-L indicate high-low, low-high, high-high and low-low nutrient treatments, respectively. The error bars represent standard errors (SE), and bars with different lower-case and upper-case letters indicate significant difference among clipping levels for ramets with intact and severed rhizome, respectively (*p* < 0.05). *—significant difference between ramets with IR vs. SR at the same clipping level.

**Figure 3 ijerph-17-02724-f003:**
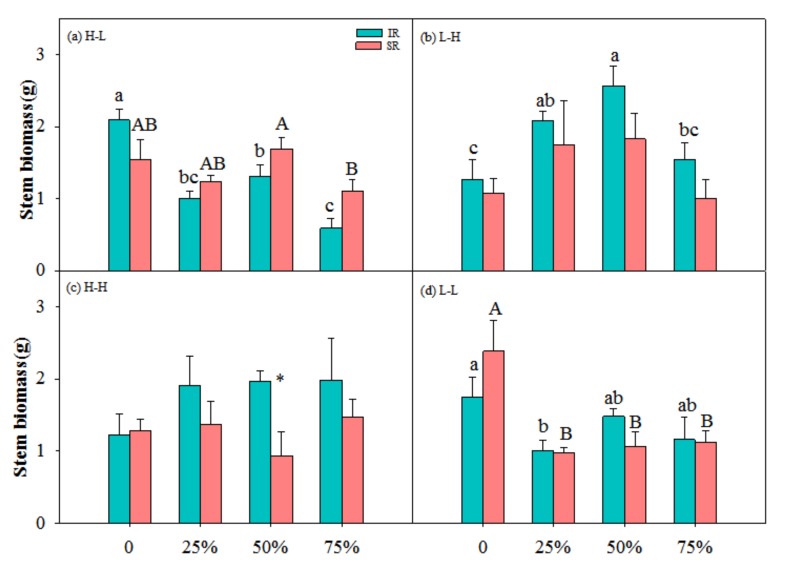
Stem biomass of ramets with intact rhizome (IR) and severed rhizome (SR) with increasing clipping levels in different nutrient treatments. (**a**) H-L, (**b**) L-H, (**c**) H-H and (**d**) L-L indicate high-low, low-high, high-high and low-low nutrient treatments, respectively. The error bars represent standard errors (SE), and bars with different lower-case and upper-case letters indicate significant difference among clipping levels for ramets with intact and severed rhizome, respectively (*p* < 0.05). *—significant difference between ramets with IR vs. SR at the same clipping level.

**Figure 4 ijerph-17-02724-f004:**
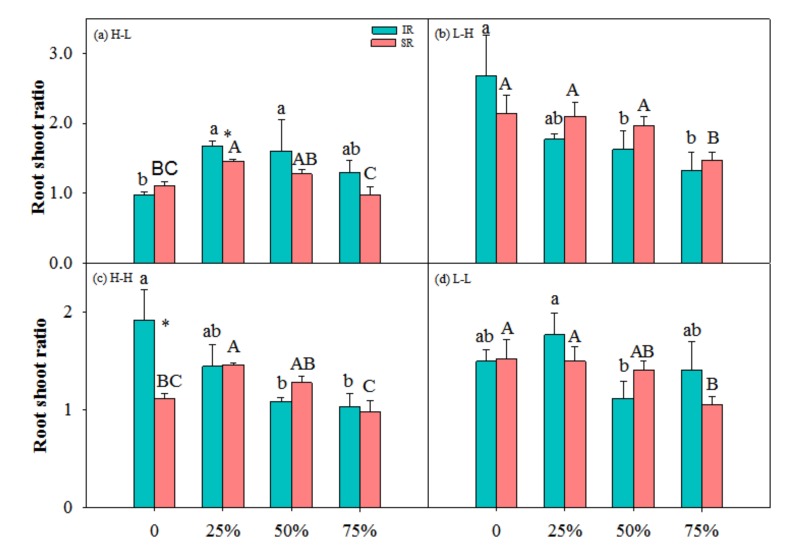
Root shoot biomass ratio of ramets with intact rhizome (IR) and severed rhizome (SR) with increasing clipping levels in different nutrient treatments. (**a**) H-L, (**b**) L-H, (**c**) H-H and (**d**) L-L indicate high-low, low-high, high-high and low-low nutrient treatments, respectively. The error bars represent standard errors (SE), and bars with different lower-case and upper-case letters indicate significant difference among clipping levels for ramets with intact and severed rhizome, respectively (*p* < 0.05). *—significant difference between ramets with IR vs. SR at the same clipping level.

**Figure 5 ijerph-17-02724-f005:**
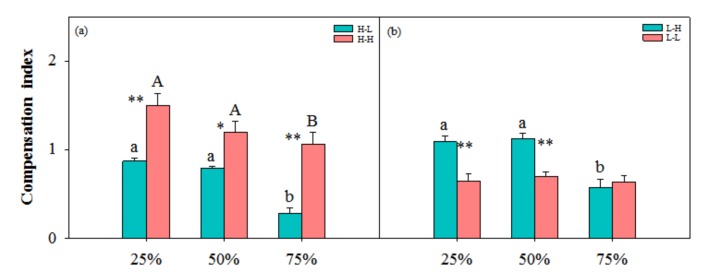
Compensation index with increasing clipping levels in different nutrient treatments. (**a**) H-L and H-H and (**b**) L-H and L-L indicate high-low, low-high, high-high and low-low nutrient treatments, respectively. The error bars represent standard errors (SE), and bars with different lower-case and upper-case letters indicate significant difference among clipping levels for ramets with different nutrient treatments (*p* < 0.05). *—significant difference between high-low vs. high-high or between low-high vs. low-low at the same clipping level.

**Figure 6 ijerph-17-02724-f006:**
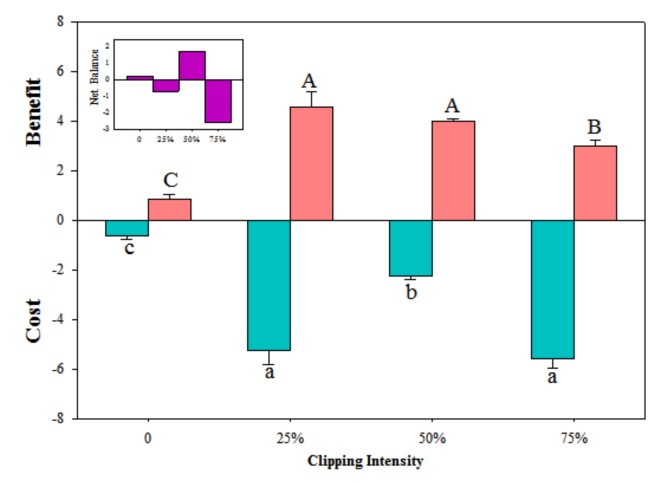
Benefit and cost of physiological integration with increasing clipping levels. The error bars represent standard errors (SE), and bars with different lower-case and upper-case letters indicate significant difference in cost and benefit, respectively, at different clipping levels (*p* < 0.05).

**Table 1 ijerph-17-02724-t001:** Results (*F* and *p* values) of three-way ANOVA on the effects of physiological integration, clipping, nutrient and their interactions on total biomass, leaf biomass, stem biomass and root shoot biomass ratio (R/S).

Treatment		TB (g)		LB (g)		SB (g)		R/S	
	df	*F*	*p*	*F*	*p*	*F*	*p*	*F*	*p*
Integration (I)	1	8.709	0.004 **	3.331	0.070	4.020	0.047 *	1.941	0.166
Clipping (C)	3	5.725	0.001 **	2.486	0.064	2.772	0.044 *	2.278	0.083
Nutrient (N)	3	1.495	0.219	2.033	0.113	2.614	0.054	1.074	0.363
I × C	3	1.290	0.210	3.133	0.028 *	0.834	0.477	3.339	0.022 *
I × N	3	4.314	0.006 **	4.005	0.009 **	3.107	0.029 *	1.891	0.134
C × N	9	3.903	0.000 **	4.554	0.000 **	5.069	0.000 ***	1.700	0.096
I × C × N	9	0.462	0.898	1.401	0.194	1.282	0.253	2.752	0.006 **

Note: * *p* ≤ 0.05; ** *p* ≤ 0.01; *** *p* ≤ 0.001.

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
