# Peer review of "The Intensity of Simulated Grazing Modifies Costs and Benefits of Physiological Integration in a Rhizomatous Clonal Plant"

_ijerph, 2020, doi:10.3390/ijerph17082724_

Round 1

Reviewer 1 Report

Comments on the manuscript: "The intensity of simulated grazing modifies costs and benefits of physiological integration in a rhizomatous clonal plant by Jushan Liu, Chen Chen, Yao Pan, Ying Gao

General comments

The paper is a good experiment about a clonal grass species, Leymus chinensis, about the effect of different nutrient supply, clipping intensity and ramet integration.

I think, it is generally well written manuscript (mc), however there is a lack of some background information in methods part, which should be completed. I suggest minor revision.

Detailed comments

Abstract

Introduction

Line 44: Please, start the sentence with capital letter

Line 76: Is the measurement of benefit and cost well developed as methods? Please, provide some reference, which shows, that the measurement of benefits and cost an accepted method.

Line 86: In this sentence (“In natural habitats, it consists mainly of sexual and asexual reproduction (rhizome growth).”) the “consist” is quite strange. I suggest to overthink it. Please, you another word for it, e.g. it use or apply.

Methods

Line 102: What kind of sand did you used? Acidic or calcareous? Regional or not? Please, indicate it!

Line 108: Where did you store the pots? Inside or outside - I mean the greenhouse was open or not? If inside, what was the environment conditions there? If outside, did you give extra water? If you give fertilizer did it mean that the plants get extra water? Please, give more detail about the environmental conditions.

Line 112: How did you ensured, that you clipped the 0%, 25%, 50% or 75% of shoot (biomass/height? It could be quite problematic to conduct it well. Please, clarify the details!

Line 116: Was the 24 hours enough? Usually the drying needs 2 days. Did you checked somehow that by this species 24 hours enough to achieved the stabile weight (that is the weight of the sample no longer changes)?

Line 118: shoot means: leaves+stems? Please, clarify it.

Line 121: As I know, divide by something, not to! – In this case.

Line 124: Please, give the exact definition, e.g. Benefits was calculated as the difference between total biomass of LH and total biomass of LL treatment. How did you treated the 5 replicates during his calculation? Did you calculated the mean total biomass of the 5 LH treatment and the subtracted the mean of the other treatment? Please, indicate it!

Line 127: Did you checked the assumptions of Anova? Please indicate it and declare of the results.

Line 129: The Tukey Kramer (or in case of equal sample size Tukey HDS) contain also the correction for the multiple testing, so you do not have to make extra Bonferroni correction on p values. Please, check it and correct it. Why did you used post -hoc test only in case of total biomass and R/S ratio? If you used post-hoc also in case of leaf and shoot biomass, too, than please rewrite the sentence and indicate it clearly.

Result

Line 145 “Physiological integration and clipping had 145 significant effects on total and stem biomass, but nutrient had no effects on any biomass variable” – if there is significant triple interaction or significant double interaction, than there is no reason to interpret the effect of one factor, because it has an effect with interaction of others, not individually. I suggest to start the description with the triple interaction and explain it in detail, then with double interaction. And explain separately what is the result in case of total biomass, leaf biomass, etc. Now it is a little bit hard to follow the result.

Discussion

Line 222: The English language of the first sentence is strange.. I miss something from this sentence, as it is hardly understandable.

Line 293: I miss the reference (which previous study?)

Line 318: I think, YZ should be also author, as he/she did the data collection, which is a very important part of the study.

Author Response

Response to Reviewer 1 Comments

Point 1: Line 44: Please, start the sentence with capital letter

Response: Thanks, revised as you suggested (Line 45).

Point 2: Line 76: Is the measurement of benefit and cost well developed as methods? Please, provide some reference, which shows, that the measurement of benefits and cost an accepted method.

Response: Yes, this method was used in our published paper (Gao et al. 2014) and other papers. The method was clearly clarified in Methods section and we add other references (Line 134).

Point 3: Line 86: In this sentence (“In natural habitats, it consists mainly of sexual and asexual reproduction (rhizome growth).”) the “consist” is quite strange. I suggest to overthink it. Please, you another word for it, e.g. it use or apply.

Response: Thanks, changed as you suggested (Line 90).

Point 4: Line 102: What kind of sand did you used? Acidic or calcareous? Regional or not? Please, indicate it!

Response: Thanks, some information about the sand was added into the revised MS (Line 106).

Point 5: Line 108: Where did you store the pots? Inside or outside - I mean the greenhouse was open or not? If inside, what was the environment conditions there? If outside, did you give extra water? If you give fertilizer did it mean that the plants get extra water? Please, give more detail about the environmental conditions.

Response: Yes, the plants were kept in an open greenhouse, which has been further clarified in the revised MS (Line 118-120).

Point 6: Line 112: How did you ensured, that you clipped the 0%, 25%, 50% or 75% of shoot (biomass/height? It could be quite problematic to conduct it well. Please, clarify the details!

Response: The clipping treatments were performed based on plant height and it was easily conducted with a ruler (Line 118).

Point 7: Line 116: Was the 24 hours enough? Usually the drying needs 2 days. Did you checked somehow that by this species 24 hours enough to achieved the stabile weight (that is the weight of the sample no longer changes)?

Response: It was 48 hours rather than 24 hours, which has been corrected (Line 124).

Point 8: Line 118: shoot means: leaves+stems? Please, clarify it.

Response: Clarified as suggested (Line 125-126).

Point 9: Line 121: As I know, divide by something, not to! – In this case.

Response: Thanks, corrected (Line 129).

Point 10: Line 124: Please, give the exact definition, e.g. Benefits was calculated as the difference between total biomass of LH and total biomass of LL treatment. How did you treated the 5 replicates during his calculation? Did you calculated the mean total biomass of the 5 LH treatment and the subtracted the mean of the other treatment? Please, indicate it!

Response: The indexes were calculated for each pot rather than the mean total biomass of the 5 replicates, which has been clarified in the revised MS (Line 134-135).

Point 11: Line 127: Did you checked the assumptions of Anova? Please indicate it and declare of the results.

Response: Yes, we checked and clarified (Line 145-147).

Point 12: Line 129: The Tukey Kramer (or in case of equal sample size Tukey HDS) contain also the correction for the multiple testing, so you do not have to make extra Bonferroni correction on p values. Please, check it and correct it. Why did you used post -hoc test only in case of total biomass and R/S ratio? If you used post-hoc also in case of leaf and shoot biomass, too, than please rewrite the sentence and indicate it clearly.

Point 13: Thanks, revised all the results only with Tukey-Kramer test and remove the section about Bonferroni correction.

  Yes, post -hoc test was used for all the variables, which has been clarified (Line 139-140)

Point 14: Line 145 “Physiological integration and clipping had 145 significant effects on total and stem biomass, but nutrient had no effects on any biomass variable” – if there is significant triple interaction or significant double interaction, than there is no reason to interpret the effect of one factor, because it has an effect with interaction of others, not individually. I suggest to start the description with the triple interaction and explain it in detail, then with double interaction. And explain separately what is the result in case of total biomass, leaf biomass, etc. Now it is a little bit hard to follow the result.

Response: Revised as suggested (Line 151-154).

Point 15: Line 222: The English language of the first sentence is strange.. I miss something from this sentence, as it is hardly understandable.

Response: Corrected (Line 228).

Comment: Line 293: I miss the reference (which previous study?)

Point 16: Sorry, we made a mistake and have corrected it (Line 299).

Comment: Line 318: I think, YZ should be also author, as he/she did the data collection, which is a very important part of the study.

Response: Thanks, we have added YZ into authors.

Reviewer 2 Report

Reviewing of the paper “The intensity of simulated grazing modifies costs and benefits of physiological integration in a rhizomatous clonal plant

The subject is of interest and in the scope of journal. This paper presents interesting material carried out on the aim of this work was to report results of a study that crossed nutrient, clipping and rhizome connection treatments to examine the role of physiological integration in compensatory growth of a clonal species Leymus chinensis. The paper is well structured and includes original data that maybe of interest for the readers of International Journal of Environmental Research and Public Health. I recommend this paper – after correction.

Comments:

- comments were marked in the text of manuscript

Author Response

Response to Reviewer 2 Comments

Point 1: Line 29, add “Leymus chinensis” into keywords

Response 1: added as suggested (Line 29).

Point 2: Line 75, explain the abbreviation, because L. chinensis may refer to other species.

Response 2: Explained as suggested (Line 76).

Point 3: Line 82: add “altitude”.

Response 3: added as suggested (Line 86).

Point 4: Line 94: add “year”.

Response 4: added as suggested (Line 98).

Reviewer 3 Report

The manuscript by Liu et al., investigates the role of physiological integration in compensatory growth of Leymus chinensis, with the aim to understand how simulated herbivory affects the physiological integration.

The biggest limitation that I’ve found by reviewing the manuscript, is the absence of the evaluation of biochemical parameters related to the health status of the plant (e.g. chemical parameters such as the chlorophyll content). Why the authors have not considered these parameters?

The manuscript is generally clear and well written. It describes a well-planned study. Although there is some lack in introduction, M&M and reference section, the methods are suited to address the questions and supported by a good statistical analysis. The conclusions are well supported by the results. The authors make clear the intended practical application.  

However, during the reviewing procedure, some mistakes were found:

INTRODUCTION: the authors use Leymus chinensis (Trin.) Tzvel. as model plant for the experimental studies of this work. However, Leymus chinensis is not mentioned in the introduction. General information about the plant (distribution, ecological importance, etc.) might be added in this section together to the related references. For these reasons, I suggest to the authors to move the part reported in Materials and Methods (LINE 85-92) in the Introduction section (LINE 75 before Our aims…)

LINE 38: there is no "dot" at the end of the first sentence

LINE 44: capital letter at the beginning of the sentence

LINE 46: “It makes it possible to forage limiting resources” considering a rewording of the sentence

Section 2.2. Has the experimental design used for this work ever been used previously by the authors? Has a similar approach been used by other researchers? If yes, please insert references.

Section 2.3. Please, report the commercial name, type (analytical or technical) and the brand of the scale used for the collection of the weight data.

How was the length of the roots measured? Did the authors use any software based on images? Please, integrate all these missing information.

RESULTS: some errors occur in the style and position of the table and figure legends. MDPI requires that the table captions are reported before the table to which it refers. On the other hand, the captions of the figures must be reported after the figure to which they refer. If the authors used Microsoft Word with the official template of IJERPH, I suggest standardizing the style of the legends using the MDPI pre-set default styles (MDPI_4.1 for table caption, and MPDI_5.1 for figure legends).

The quality of the figures may be improved: IJERPH allows to use colour figures with no additional cost. Please, consider submitting a better quality version of the figures.

Figure 1 caption: LINE 165 “(a) H-L, (b)L-H, (c)H-H and (d)” there is no space between the panel letter and the description.

Figure 2 caption: LINE 173, see previous comment

Y axis figure 2: It should be leaf biomass not leaves biomass

Figure 3 caption: LINE 180, see previous comment

Figure 4 caption: LINE 195, see previous comment

LINE 253: Linaria vulgaris must be written in italics

REFERENCES: excluding two articles dated in 2017, the most recent article is from 2014. I suggest to add some recent references, ranging between 2018 and 2020, if available.

Author Response

Response to Reviewer 3 Comments

Point 1: The biggest limitation that I’ve found by reviewing the manuscript, is the absence of the evaluation of biochemical parameters related to the health status of the plant (e.g. chemical parameters such as the chlorophyll content). Why the authors have not considered these parameters?

Response 1: Thanks for your advice. The costs and benefits of physiological integration are usually calculated based on fitness and biomass. Chemical parameters will be evaluated in further study.

Point 2: INTRODUCTION: the authors use Leymus chinensis (Trin.) Tzvel. as model plant for the experimental studies of this work. However, Leymus chinensis is not mentioned in the introduction. General information about the plant (distribution, ecological importance, etc.) might be added in this section together to the related references. For these reasons, I suggest to the authors to move the part reported in Materials and Methods (LINE 85-92) in the Introduction section (LINE 75 before Our aims…)

Response 2: Thanks, we have added more introduction of this species into this section (Line 76-79).

Point 3: LINE 38: there is no "dot" at the end of the first sentence

Response 3: Thanks, added (Line 39).

Point 4: LINE 44: capital letter at the beginning of the sentence

Response 4: Thanks, changed in the revised MS (Line 45).

Point 5: LINE 46: “It makes it possible to forage limiting resources” considering a rewording of the sentence

Response 5: Revised as suggested (Line 47).

Point 6: Section 2.2. Has the experimental design used for this work ever been used previously by the authors? Has a similar approach been used by other researchers? If yes, please insert references.

Response 6: Yes, the experimental design has been widely used and some references has been added (Line 113).

Point 7: Section 2.3. Please, report the commercial name, type (analytical or technical) and the brand of the scale used for the collection of the weight data. How was the length of the roots measured? Did the authors use any software based on images? Please, integrate all these missing information.

Response 7: Thanks, information about the balance for biomass was added into the revised MS (Line 123-124).

  We did not measure root length and only root biomass was used in this study.

Point 8: RESULTS: some errors occur in the style and position of the table and figure legends. MDPI requires that the table captions are reported before the table to which it refers. On the other hand, the captions of the figures must be reported after the figure to which they refer. If the authors used Microsoft Word with the official template of IJERPH, I suggest standardizing the style of the legends using the MDPI pre-set default styles (MDPI_4.1 for table caption, and MPDI_5.1 for figure legends).

Response 8: Thanks, revised as you suggested (Line 163-165).

Point 9: The quality of the figures may be improved: IJERPH allows to use colour figures with no additional cost. Please, consider submitting a better quality version of the figures.

Response 9: Thanks, the figures have been replaced with colour figures in higher quality.

Point 10: Figure 1 caption: LINE 165 “(a) H-L, (b)L-H, (c)H-H and (d)” there is no space between the panel letter and the description.

Response 10: Thanks, changed as you suggested (Line 171).

Point 11: Figure 2 caption: LINE 173, see previous comment

Response 11: Changed (Line 179).

Point 12: Y axis figure 2: It should be leaf biomass not leaves biomass

Response 12: Changed (Figure 2).

Point 13: Figure 3 caption: LINE 180, see previous comment

Response 13: Changed (Line 186).

Point 14: Figure 4 caption: LINE 195, see previous comment

Response 14: Changed (Line 201).

Point 15: LINE 253: Linaria vulgaris must be written in italics

Response 15: Thanks, changed as you suggested (Line 259).

Point 16: REFERENCES: excluding two articles dated in 2017, the most recent article is from 2014. I suggest to add some recent references, ranging between 2018 and 2020, if available.

Response 16: Thanks, some recent published articles have been added into the citations (Line 36, 43, 58).